# Exclusive Breastfeeding and Childhood Morbidity: A Narrative Review

**DOI:** 10.3390/ijerph192214804

**Published:** 2022-11-10

**Authors:** Saldana Hossain, Seema Mihrshahi

**Affiliations:** Department of Health Sciences, Faculty of Medicine, Health and Human Sciences, Macquarie University, Sydney, NSW 2109, Australia

**Keywords:** exclusive breastfeeding, diarrhea, acute respiratory infection, gastrointestinal infection, respiratory infection

## Abstract

Globally, diarrheal diseases and acute respiratory infections are the leading causes of morbidity and mortality in children under 5 years old. The benefits of exclusive breastfeeding in reducing the risk of gastrointestinal and respiratory infections are well documented. Optimal breastfeeding may potentially save the lives of about 800,000 children in low-income settings. Despite the evidence, around 63% of infants from birth to 6 months are not exclusively breastfed worldwide. We searched the literature published between 2010 and 2022 in Medline, Embase, and Scopus on the association between exclusive breastfeeding and infectious diseases. We selected and reviewed 70 relevant studies. Our findings expand and confirm the positive association between exclusive breastfeeding and reduced risk of a number of gastrointestinal, respiratory, and other infections in 60 out of 70 studies observed in both low- and high-income settings. Several studies analyzing exclusive breastfeeding duration reported that a longer exclusive breastfeeding duration is protective against many infectious diseases. This review also reported a lack of standardized definition for measuring exclusive breastfeeding in many studies. Overall, the results highlight the benefits of exclusive breastfeeding in many studies and suggests reporting exclusive breastfeeding in future studies using a consistent definition to enable better monitoring of exclusive breastfeeding rates.

## 1. Introduction

Infectious diseases are a leading cause of deaths in children below the age of 5 worldwide [1]. The most common infectious diseases affecting young children are diarrhea and acute respiratory infections [1]. Despite enormous efforts in the prevention of diarrheal morbidity and mortality, diarrhea is the second largest cause of mortality among children under 5 years old, and almost 1.7 billion children are experiencing diarrheal occurrences worldwide [2]. Globally, diarrhea leads to 525,000 deaths in children per year [2]. Furthermore, acute respiratory infection (ARI) is responsible for almost 20% of all deaths in children [3,4]. In 2016, almost 653,000 deaths were attributed to lower ARIs in children below 5 years old [3].

Research suggests that exclusive breastfeeding (EBF) is an important protective factor against infant morbidity and mortality from respiratory infections and gastrointestinal tract infections [5]. The World Health Organization (WHO) recommends EBF until the age of 6 months, and then continuation of breastfeeding along with complementary feeding until 2 years or more [6]. The WHO defines EBF as a practice, whereby the baby receives only breast milk in the first 6 months of life with no other foods or liquids, except for medications, vitamin and mineral supplements, and oral rehydration salts [6]. Approximately 11.6% of all deaths in children have been attributed to suboptimum breastfeeding in 2011 [7]. Despite the evidence supporting EBF, about 63% of babies aged less than 6 months in low- and middle-income countries (LMICs) do not receive EBF [8]. EBF has the potential to avert roughly 12% of deaths in children below 5 years old in LMICs [6]. The utilization of breastfeeding promotion programs and policies may be able the save the lives of millions of children, particularly in resource-limited areas where proper child nutrition and readily obtainable pediatric health services are not easily available [5,7]. However, suboptimum breastfeeding is an issue in both LMICs and high-income countries. High-income nations have a shorter breastfeeding duration compared to LMICs [8]. In 2012, the WHO adopted its Comprehensive Implementation Plan on Maternal, Infant, and Young Child Nutrition, which set six targets to lower mortality and mortality associated with nutrition [7]. One of the six objectives was to increase the rates of EBF during the first 6 months of life to 50% by 2025, which signifies the urgent need to address this issue [7,8]. A meta-analysis suggested that since 1993, EBF rates have increased by around 0.5 percentage points annually, reaching 35% in the year 2013 [8]. The meta-analysis indicates that an increase of greater than one percentage point is required every year to reach the target [8].

Thus, the purpose of this review is to critically analyze the published literature on the relationship between EBF and childhood morbidity, as very few studies have reviewed the current literature on EBF and a wide range of infections. The results of this review will build on an existing systematic review of breastfeeding and the risk of diarrhea morbidity in 2011, additionally investigate other infectious diseases such as ARI and provide recommendations for further research in this area [9].

## 2. Materials and Methods

### 2.1. Search Strategy

This review was undertaken by searching the following databases: Embase (OVID), Scopus, and Medline (OVID). Keywords and terms utilized in the searches included exclusive breastfeeding, morbidity, diarrhea, acute respiratory infection, fever, cough, and infectious diseases. Appropriate Boolean operators, truncations, and search features such as subject headings were used and adapted according to the database. The search was limited to papers published from January 2010 up to April 2022. Full details of the search strategy can be found in Appendix A.

### 2.2. Eligibility Criteria

The inclusion and exclusion criteria of studies are shown in Table 1. 

### 2.3. Selection of Studies

The articles found by using the search strategy were all exported to Endnote X9 for further screening. After duplicates were removed, as shown in Figure 1, a total of 859 studies were retrieved. After reviewing titles and abstracts, 70 papers were included in the review (Table 2). The quality assessment of each study involved an examination of the definitions of EBF and morbidity outcomes, study design, number of participants, and length of recall for the infant feeding practice. Studies were excluded if the children were older than 5 years of age at the time of interview/questionnaire as very long recall periods may result in inaccurate information about breastfeeding status. For example, a cohort study in Thailand involved 322 participants where it was found that children who were EBF for less than 6 months had lower odds of pneumonia [10]. However, the age of subjects ranged from 1 month to 15 years old, leading to a large recall length variability; therefore, they were excluded. A narrative review was conducted rather than a meta-analysis due to the heterogeneity of the definitions of exposures and morbidity outcomes, variability in the study methodologies, differing study settings, and categories of breastfeeding duration. Furthermore, this review assigned EBF status according to each author’s assigned category in the included studies, not the category based on the WHO’s definition of feeding practice, which further makes it unsuitable for a meta-analysis. Studies that provided no definition for EBF and EBF for infants beyond 6 months of age were also included.

## 3. Results

### 3.1. Study Characteristics 

Among the 70 included studies, there were six meta-analyses, 28 cross-sectional studies, 25 cohort studies, and 11 case–control studies, as detailed in Table 2. The sample size for each of the studies ranged from 116 to 502,948 participants. Data from these studies were mostly collected in developing countries (49 out of 70). A total of 42 out of the 49 identified studies in developing countries found that EBF was associated with a significant reduced risk of infection. A total of 14 out of the 17 papers in high-income countries reported decreased infectious disease risk in the EBF group. All four studies conducted in both LMICs and high-income countries (multi-country) indicated lower infection risk in the EBF group. Almost all studies that analyzed the duration of EBF reported the protective effects of a longer EBF duration. A total of 16 studies used the WHO definition of EBF for describing the explanatory variable, and two studies provided a slight variant of the WHO definition of EBF [56,75]. A total of 32 studies provided no specific definition for EBF. In terms of the methods of reporting EBF, 17 papers utilized the 24 h recall measure, and some papers used weekly or twice a week home visits, 2 week recall, and recall since birth measures. Most of the studies utilized self-administered questionnaires or surveys to measure EBF practices. 

### 3.2. Diarrhea and Other Related Gastrointestinal Infections 

Out of the 28 studies that included diarrhea as the outcome variable, nearly all studies (*n* = 27; one meta-analysis, 19 cross-sectional, five cohort, and two case–control studies) found a statistically significant decrease in the risk of diarrhea in children who were EBF [14,18,24,27,31,39,42,43,53,55,56,58,60,62,63,65,66,67,69,70,71,72,74,75,76,77,78]. Only one paper did not find a statistically significant association, and this analysis looked at diarrhea and ARI comorbidity [68]. Of the four studies with rotavirus as an outcome, two studies found a reduction in rotavirus infection and rotavirus diarrhea in children who were EBF, and two studies found no significant relationship [15,16,23,54]. All 11 papers that investigated ETEC (*n* = 2), EAEC (*n* = 1), *giardia* (*n* = 1), *Campylobacter* infections (*n* = 2), GI infections (*n* = 4), and enteric infections (*n* = 1) reported the protective effects of EBF [17,19,20,22,25,26,28,32,33,38,41]. 

### 3.3. ARI and Other Respiratory Infections 

A total of 17 articles examined the relationship between ARI/acute lower respiratory infections and EBF, of which the majority of the articles noted a decrease in ARI in the EBF group (*n* = 13) [13,14,24,36,46,53,55,59,63,69,73,74]. A total of three papers found no significant relationship between EBF and ARI [50,68,77]. One paper from Germany reported that a longer duration of EBF led to a higher risk of ARI [37]. All studies assessing lower respiratory tract infections (*n* = 6), pneumonia (*n* = 5), bronchiolitis (*n* = 1), and broad respiratory/chest infections (*n* = 5) reported a protective association with EBF [11,18,21,28,32,33,34,35,39,40,41,48,49,50,56,57,79]. A few studies found that there was no significant association between EBF and tuberculosis (*n* = 1), upper respiratory tract infection (*n* = 1), and pertussis (*n* = 1) [44,52,61]. Three studies examining upper respiratory tract infection and pertussis found EBF as a protective factor [28,32,47]. One publication of a study conducted in Australia reported that EBF lead to increased risk of human rhinovirus infection; however, the sample size (*n* = 167) was small [29]. 

### 3.4. Fever and Infections

All papers that analyzed fever as the outcome factor reported a decrease in the risk of fever in children who were EBF (*n* = 5; 4 cross-sectional, 1 cohort) [21,66,67,69,74]. All articles that examined urinary tract infection (*n* = 1), sepsis (*n* = 1), acute otitis media (*n* = 3), HFMD infection (*n* = 1), and broad category of infections (*n* = 2) reported a reduced risk of these conditions in children who were EBF [12,14,30,33,36,45,64,80]. 

## 4. Discussion

### 4.1. Main Findings

This review updates and confirms the strong protective effects of EBF against childhood morbidity, which reaffirms the previous systematic reviews published by Lamberti et al. [9,81] and others [5,82,83]. It further expands upon the evidence base established by Lamberti et al. [9,81], who predominantly looked at diarrhea and pneumonia. This review highlights the beneficial association between EBF and the decreased risk of a larger number of gastrointestinal, respiratory, and other infections in both LMICs and high-income countries. Only 10 of these 70 studies found either no significant association or an increased risk of infections due to EBF. However, most of these results may be attributed to their small sample size of fewer than 300 participants [29,44,51,52,54,61]. Almost all studies in this review that looked at diarrhea and other gastrointestinal infections suggested the protective effects of EBF, which is in agreement with the paper by Lamberti et al. [9], where diarrhea mortality was greater in infants who were not breastfed relative to those who were EBF (RR = 10.52). In addition, most of the identified papers that examined the duration of EBF found that babies who were EBF for a longer duration experienced lower morbidity from many infectious diseases such as UTI, expanding on the review by Kramer et al. [5] that only noted its impact on gastrointestinal infections. 

Another important finding from our review was the use of inconsistent definition for EBF. For instance, a study in Japan defined EBF as the infant’s consumption of breast milk as the only source of milk but may or may not include other foods and liquids [38]. There is a dearth of literature discussing the large variations in definitions of EBF used. Khanal et al. [84] and Binns et al. [85] reported this discrepancy in definition used to measure EBF in studies in Nepal and Australia. A few studies included in our review claimed to adopt the WHO definition of EBF; however, it was noted that the definition was not used accurately [56,75,85]. Consistency in the definition of EBF is crucial to appropriately detect and compare the changes and trends in EBF prevalence over time, across countries and within countries for development and progress of public health policies and interventions [84,86]. Additionally, there was also some variation in the definitions of morbidity outcomes. For example, Diallo et al. [31] in USA defined diarrhea as experiencing at least one episode of diarrhea, whereas the DHS studies defined this as three or more episodes of diarrhea per day [70]. Furthermore, surveys and questionnaires given to participating mothers was the most common measurement tool for collecting information on breastfeeding practices. 

This review supports the current WHO recommendation of EBF for the first six months of life. Breastfeeding is influenced by sociocultural and economic factors, maternal factors such as age, education, work environment, birth, and postpartum circumstances, and perspectives on EBF; hence, initiatives must be taken according to the context of LMIC or high-income setting [83,87]. Breastfeeding requires support from families, communities, healthcare professionals, and the government; mothers are not responsible for this alone [74]. Sufficient breastfeeding counseling, lactation centers, education programs for both mothers and healthcare staff and other effective and integrated health policy interventions are vital for mothers in order to initiate and sustain optimal EBF practice [83]. 

### 4.2. Biological Plausibility 

Numerous components in breast milk provide the biological basis that may explain its protective effects against gastrointestinal and respiratory diseases. It contains several bioactive elements that add to its total immunological activity, including antibodies such as immunoglobulin A (IgA), prebiotic oligosaccharides, lactoferrin, nonspecific anti-infective agents, lymphocytes, leucocytes, probiotics, and other immune cells and beneficial microbes [87,88]. Some of these beneficial components such as secretory antibodies, oligosaccharides, hormones, stem cells, enzymes, and immune cells are not present in formula milk [89]. Compared to formula milk, human milk also has a dynamic composition that varies over time to accommodate to the changing needs of the developing infant [89]. These elements offer the mechanism for infection prevention, especially during the first few months of life and during the breastfeeding phase [87,88]. Research shows that IgA stops the attachment of viruses and bacteria to the mucosal epithelial cells, which might lead to infections [88,89]. IgA antibodies defend against microbes that cause gastrointestinal infections such as *Giardia*, ETEC, and *campylobacter* [88,89]. Lactoferrin works as an antimicrobial agent, killing pathogens [88]. Oligosaccharides may block the attachment of pathogens such as pneumococci to the mucous membrane, thereby hindering the formation of respiratory and gastrointestinal infections [88,89]. Moreover, there is a possibility that breastmilk may cover the nasopharyngeal mucosa, thus safeguarding against the transmission of pathogens that cause respiratory diseases [90,91]. 

Another explanation is that babies who are not breastfed are more exposed to diarrheal-causing pathogens through feeding bottles, nonhuman milk, or other given foods to them, which prevents babies from receiving sufficient nutrition and, therefore, nonspecific immunity [84,92]. For instance, a study reported that typical weaning foods were infected with microbials that might induce gastrointestinal diseases in Gambian babies [92]. 

There is also evidence that breastfeeding may reduce the risk of sudden infant death syndrome (SIDS) with plausible biological mechanisms for this effect [93,94]. Cytokines and immunoglobulins may safeguard babies during the period when SIDS risk is greatest (2–4 months of age) [94]. Many babies who died from SIDS had a mild infection in the days before death; however, this is insufficient to account for the deaths [94]. It is probable that these infections may produce proinflammatory cytokines that could result in complications [94].

### 4.3. Strengths and Limitations

The current review was limited as we did not examine randomized controlled trials which would have provided high-quality evidence on the association between EBF and morbidity [87]. It is unethical to randomly assign infants to EBF; therefore, observational studies and reviews were examined [87]. However, it is ethically appropriate to promote EBF in communities compared to providing no EBF promotion [95]. For instance, in Bangladesh, the Shishu Pushti trial randomly allocated pregnant women to peer counseling for infant feeding practices (intervention) or to the control group [96]. Furthermore, most of the studies in this review did not account for reverse causality bias; that is, EBF might have been stopped or changed due to an occurrence of ARI, gastrointestinal, or other morbidity outcomes [9]. This may result in an overestimation of the protective effects of EBF against morbidity since the prevalence of EBF is underestimated in infants who develop the morbidity outcome [95]. Another limitation is that some of the studies in this review did not adjust for confounding factors such as poverty and maternal education [95]. Poverty has been associated with prolonged breastfeeding duration in LMICs; therefore, this may result in an inaccurate estimation of the protective effect of EBF [9,95]. Self-selecting mothers who breastfed for a greater duration may also have led to bias, especially in high-income countries, where breastfeeding mothers are more likely to be educated and health-conscious [95]. This review may also be limited by some studies with a longer length of recall, where mothers who breastfed for a shorter time may be more likely to overestimate EBF duration resulting in the possibility of misclassification. [95] Nevertheless, a major strength is that we carried out an inclusive and comprehensive approach to include a large number of studies with a wide geographic diversity from both LMICs and high-income countries, which makes it more generalizable to different populations. 

## 5. Conclusions

Overall, our review confirms the results of previous reviews on the beneficial association between EBF and reduced childhood morbidity including gastrointestinal, respiratory, and other infections and fever in both LMICs and high-income countries. The measurement of EBF using a standardized and strict terminology is also critical in future research studies for reporting EBF levels and precisely comprehending the health benefits [84,86]. 

## Figures and Tables

**Figure 1 ijerph-19-14804-f001:**
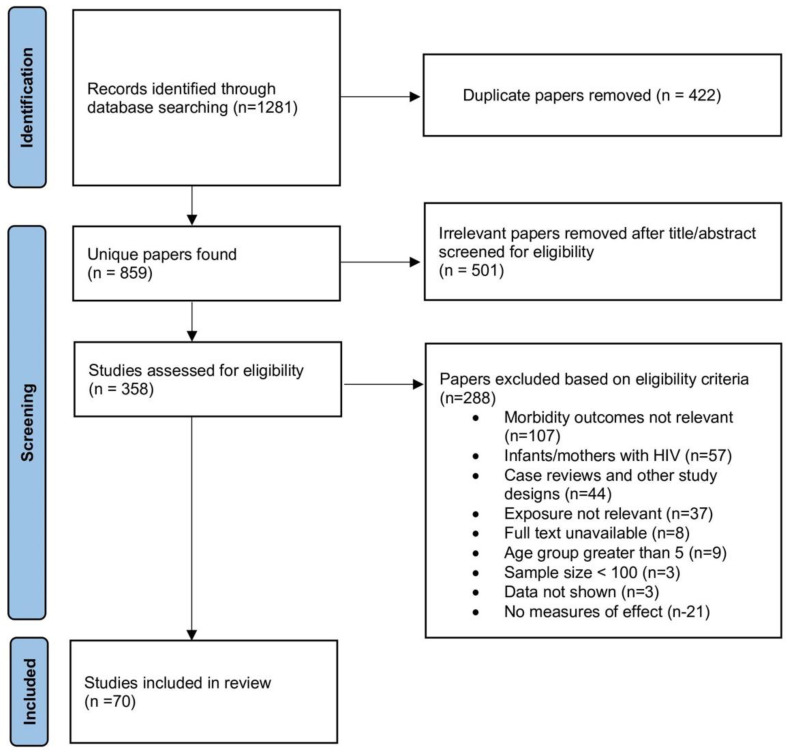
Flow diagram of study selection process.

**Table 1 ijerph-19-14804-t001:** Eligibility criteria of reviewed studies.

Inclusion Criteria	Exclusion Criteria
Studies that measured the association between EBF and childhood morbidity	Child age older than 5 years
Child age less than 5 years	Studies on HIV-exposed babies and children (reason: altered immune status of these children)
Study outcome of interest was diarrhea, respiratory tract infections, fever, and other infectious diseases	Studies with outcomes such as asthma, rhinitis, COVID-19, allergies, wheezing, gut microbiota, malaria, stunting, underweight, wasting, and mother’s morbidity
Published in a peer-reviewed journal	Sample size < 100
Randomized controlled trials, observational studies, and meta-analysis	Case studies/reports, letters, conference abstracts, reviews, and nonhuman studies
Written in English language	No measurement of association/effect

**Table 2 ijerph-19-14804-t002:** Summary table of studies investigating the association between EBF and childhood infections arranged according to study design and country type (*n* = 70).

Author(s), (year)	Country	Sample Size	Dataset	Exposure Variable	Outcome Variables	Main Results
**Meta-analysi**s						
Alamneh et al. (2020) [11]	Ethiopia	4598		EBF, during first 6 months	Pneumonia	Non-EBF group had a higher risk of developing pneumonia than EBF group (OR = 2.46; 95% CI 1.35, 4.47).
Bowatte et al. (2015) [12]	Multi-country	11,349		EBF, first 6 months	Acute otitis media (AOM)	EBF was associated with lower odds of AOM (OR = 0.57; 95% CI 0.44, 0.75) than ‘ever vs. never’ BF (OR = 0.67; 95% CI 0.56, 0.80) and ‘more vs. less’ BF (OR = 0.67; 95% CI 0.59, 0.76).
Jackson et al. (2013) [13]	Multi-country	n/a		EBF, only breast milk during first 4 months	Acute lower respiratory infections	Lack of EBF was associated with severe acute lower respiratory infection (OR = 2.34; 95% CI 1.42, 3.88)
Khan et al. (2015) [14]	Multi-country	70,976		EBF, no definition	Morbidity—diarrhea, ARI, sepsis	Partially breastfed infants had a greater risk of diarrhea (RR = 2.97; 95% CI 1.38, 6.41), sepsis (RR = 3.46; 95% CI 2.41, 4.98), and ARI (RR = 1.69; 95% CI 1.08, 2.63) than EBF group.
Krawczyk et al. (2016) [15]	Multi-country	3466		EBF, no definition	Rotavirus infection	EBF decreases risk of rotavirus infection (OR = 0.62; 95% CI 0.48, 0.81).
Shen et al. (2018) [16]	LMICs	10,811		EBF, no definition	Rotavirus diarrhea	No significant relationship between case and control group. No correlation between rotavirus and EBF (OR = 0.86; 95% CI 0.63, 1.16).
**Cohort studies (developing countries)**						
Amour et al. (2016) [17]	Nepal, Peru, Pakistan, South Africa, India, Tanzania, Brazil, and Bangladesh	1892	MAL-ED study data	EBF, home surveillance twice a week categorized BF on day before as exclusive (no intake of other drinks or foods)	*Campylobacter* Infection	EBF group had decreased risk of *Campylobacter* detection than non-EBF group (RR = 0.57; 95% CI 0.47, 0.67)
Hanieh et al. (2015) [18]	Vietnam	1049	Ha Nam study data	EBF, only breast milk since birth with no other liquids or solids except medications, oral drops, and vitamins (questionnaire)	Diarrhea, pneumonia	EBF at 6 weeks since birth decreased the odds of inpatient admission for diarrhea (OR 0.37; 95% CI 0.15, 0.88) and suspected pneumonia (OR 0.39; 95% CI 0.20, 0.75)
Haque et al. (2019) [19]	Nepal, Peru, Pakistan, South Africa, India, Tanzania, Brazil, and Bangladesh	1715	MAL-ED birth cohort data	EBF, no definition	*Campylobacter* infection	Shorter duration of EBF (IRR = 0.98; 95% CI 0.95, 0.99; *p* = 0.035) was associated with *Campylobacter* infection.
Hassan et al. (2014) [20]	Egypt	348	Abu Homos study data	EBF, no definition	Diarrhea- ETEC	EBF decreased the risk of ETEC diarrhea (aRR = 0.51; 95% CI 0.3, 0.7, *p* < 0.001).
Henkle et al. (2013) [21]	Bangladesh	331	Mother’s Gift study data	EBF, only breast milk 1 week prior to each weekly assessment (interview)	Respiratory illness with fever	A significant protective effect of EBF was found for respiratory disease (aOR = 0.59; 95% CI 0.45, 0.77) for EBF group compared to non-EBF group.
Mansour et al. (2014) [22]	Egypt	348	Abu Homos study data	EBF, breast milk a sole source of nutrition (questionnaire)	ETEC diarrhea- ETEC phenotypes- ST, LT	EBF was associated with a reduction in risk of ST (IRR = 0.47; 95% CI 0.24,0.91) and LT/ST-ETEC related diarrhea (IRR = 0.19; 95% CI 0.05, 0.67).
Panda et al. (2014) [23]	India	696	South-24 Parganas study data	EBF, no definition (questionnaire)	Rotavirus diarrhea	Non-EBF group less than 6 months of age had greater odds of diarrhea than non-EBF group (OR = 2.07; 95% CI 1.1, 3.82).
Richard et al. (2018) [24]	Nepal, Peru, Pakistan, South Africa, India, Tanzania, Brazil, and Bangladesh	1731	MAL-ED study data	EBF (questionnaire)	Acute lower respiratory infection, diarrhea	A significant protective effect of EBF during first 6 months was found against diarrhea (RR = 0.39; 95% CI 0.32, 0.49 for 0 to 2 months), (RR = 0.83; 95% CI 0.75, 0.93) and acute lower respiratory infection (RR = 0.81; 95% CI 0.68, 0.98 for 3 to 5 months).
Rogawski et al. (2017) [25]	(Stated above)	2089	MAL-ED study data	EBF, no definition	*Giardia* Infection	A significant protective effect of EBF was found against *Giardia* infection (HR = 0.46; 95% CI 0.28, 0.75).
Rogawski et al. (2017) [26]	(Stated above)	2092	MAL-ED study data	EBF, no definition	Enteroaggregative *Escherichia coli* (EAEC) infection	A significant protective effect of EBF was found against (EAEC) infection (RR = 0.72; 95% CI 0.65, 0.79).
Rogawski et al. (2015) [27]	India	465	Vellore slum data	EBF, WHO definition (home visit data collection)	Diarrhea	In babies who stopped EBF prior to 6 months, antibiotic exposure was associated with higher risk of diarrhea (IRR = 1.48; 95% CI 1.23, 1.78).
**Cohort studies (developed countries)**						
Ajetunmobi et al. (2015) [28]	Scotland	502,948	Hospital record linkage	EBF, main mode of feeding on day before data collection	Hospitalization due to GI, respiratory tract infections, etc.	Higher risk of hospitalization for upper respiratory tract (HR = 1.28; 95% CI 1.17, 1.40), lower respiratory tract (HR = 1.50; 95% CI 1.41–1.59), and GI infections (HR = 1.59; 95% CI 1.47, 1.73) among formula-fed babies relative to EBF-group.
Annamalay et al. (2012) [29]	Australia	167	The Kalgoorlie Otitis Media research Project (KOMrP) data	EBF at 6 to 8 weeks postpartum (Interview)	Human rhinovirus infection	In non-Aboriginal babies, EBF was associated with a higher risk of HRV-a detection (OR = 3.08; 95% CI 1.09, 8.76).
Christensen et al. (2020) [30]	Denmark	815	Odense Child Cohort (OCC) data	EBF (WHO definition) (questionnaire)	Infections	Increased duration of EBF was associated with a decreased rate of hospitalization due to any infection (IRR: 0.88; 95% CI 0.80, 0.96; *p* = 0.006).
Diallo et al. (2020) [31]	USA	1172	Infant Feeding Practices Study II (IFPS II) data	EBF, only breast milk at every infant feeding session, without other foods (questionnaire)	Diarrhea	When compared to babies who got EBF for 3 months, the odds of diarrhea between 7–12 months greater in babies who discontinued EBF prior to 3 months (OR = 1.15; 95% CI 1.08, 1.22).
Duijts et al. (2010) [32]	Netherlands	7116	Generation R Study data	EBF, only breast milk and no other solids, fluids, or milk (questionnaire)	GI, upper respiratory and lower respiratory infections	Babies who were EBF until 4 months of age and partially subsequently had reduced risk of GI (OR = 0.41; 95% CI 0.26, 0.64), upper respiratory (OR = 0.65; 95% CI 0.51, 0.83) and lower respiratory infections (OR = 0.50; 95% CI 0.32, 0.79) till 6 months of age than babies not breastfed.
Frank et al. (2019) [33]	USA, Germany, Finland, and Sweden	6861	The Environmental Determinants of Diabetes in the Young (TEDDY) data	EBF, only breast milk with no other foods or formula before enrollment (questionnaire)	GI, respiratory infection, otitis media	The odds of respiratory infection (OR = 0.72; 95% CI 0.60, 0.87), GI infection OR = 0.45; 95% CI 0.32, 0.62), and otitis media (OR = 0.64; 95% CI 0.49, 0.84) were decreased among EBF-group compared to non-EBF group.
Gomez-Acebo et al. (2021) [34]	Spain	969	Marques de Valdecilla University Hospital data	EBF (WHO Definition) (feeding type at hospital discharge from medical records)	Bronchiolitis	EBF decreased the number of bronchiolitis episodes by 41% (IR = 0.59; 95% CI 0.46, 0.76) compared to infant formula.
Kawai et al. (2011) [35]	Japan	1796	Fukushima City data	EBF, no definition (questionnaire)	Lower respiratory tract infections	Girls who were EBF for a longer duration had a lower risk of lower respiratory infections hospitalization (HR = 0.45; 95% CI 0.19, 1.04 for 3 to 4 months) (HR = 0.38; 95% CI 0.15, 0.97 for >5 months).
Ladomenou et al. (2010) [36]	Greece	926	Crete study data	EBF, only breast milk with no other liquids, water and solids (questionnaire)	AOM, ARI	Babies who were EBF for 6 months had lower risk of AOM (OR = 0.37; 95% CI 0.13, 1.05) and ARI (OR = 0.58; 95% CI 0.36, 0.92) compared to babies not breastfed or partially breastfed.
Langer et al. (2022) [37]	Germany	782	LoewenKIDS study data	Duration of EBF, no definition (questionnaire)	ARI	EBF for less than 4 months is associated with a decreased risk of ARI compared to EBF for 4 to 6 months (RR = 0.78; 95% CI 0.69, 0.89).
Nakamura et al. (2020) [38]	Japan	31 578	Longitudinal Survey of Babies data	EBF, breast milk and no formula milk (with or without other foods or fluids) (survey)	Hospitalization due to GI infection	Late preterm babies who were EBF (OR = 1.13; 95% CI 0.36, 3.61) had lower risk of GI infection hospitalization compared to late preterm babies who were not EBF (wide CI).
Quigley et al. (2016) [39]	UK	15,809	Millennium Cohort Study data	EBF, no definition Pre-2001 WHO policy (EBF for 4–6 months, given solids without formula prior to 6 months, continuing BF at 6 months)	Diarrhea, chest infection	Babies who were EBF for less than 4 months had a greater risk of diarrhea (RR = 1.42; 95% CI 1.06, 1.92) and chest infection (RR = 1.24; 95% CI 1.05, 1.45) than pre-2001 WHO policy babies.
Rosas-Salazar et al. (2022) [40]	USA	1949	INSPIRE study data	EBF, no definition (parental report)	Lower respiratory tract infection	A significant protective effect of EBF was found against lower respiratory tract infection (OR = 0.95; 95% CI 0.91, 0.99 for each 1 month of EBF)
Videholm et al. (2021) [41]	Sweden	37,825	Uppsala County data	EBF, WHO definition (Child Healthcare Quality database)	Respiratory, enteric infection	Non-EBF group had a greater risk of respiratory infections (IRR = 2.53; 95% CI 1.51, 4.24) and enteric infections (IRR = 3.32; 95% CI 2.14, 5.14) than babies who were EBF for 6 months or more.
**Case–control studies (developing countries)**						
Baye et al. (2021) [42]	Ethiopia	357	Dessie city study data	EBF (recall period > 2 weeks before survey)	Acute diarrhea	The odds of acute diarrhea in non-EBF group during the first 6 months were higher EBF group (OR = 2.12; 95% CI 1.15, 3.70).
Delelegn et al. (2020) [43]	Ethiopia	306	Dessie referral hospital data	EBF (no definition) (questionnaire)	Acute diarrhea	Non-EBF group were more likely to experience acute diarrhea relative to EBF group (AOR = 3.32; 95% CI 1.206, 9.14).
Fahdiyani et al. (2016) [44]	Indonesia	165	Tamansari primary health care data	EBF, only breast milk during first 6 months or infant’s age during questionnaire if < 6 months	Upper respiratory tract infection (URI)	EBF was not associated with URI (OR = 0.76; 95% CI 0.38, 1.50, *p* = 0.425).
Lin et al. (2014) [45]	China	937	Guangdong Province study data	EBF, no foods or any other liquids and formula milk with exception of vitamins, minerals, and medication during first 6 months (questionnaire)	Hand, foot, and mouth disease (HFMD)	EBF was associated with decreased odds of HFMD (OR = 0.60; 95% CI 0.45, 0.79) compared to mixed feeding.
Mir et al. (2022) [46]	Pakistan	3213	Taluka Kotri study data	EBF, no definition (questionnaire)	ARI	Infants who were EBF during the first 6 months had lower odds of ARI compared to non-EBF group (OR = 0.81; 95% CI 0.69, 0.96).
Nascimento et al. [47]	Brazil	267	Metropolitan Region of Recife (RMR) data	EBF, no definition (medical records)	Pertussis-like illness	Significant protective effect of EBF during first 6 months found against pertussis-like illness (OR = 0.26; 95% CI 0.11, 0.62)
Ngocho et al. (2019) [48]	Tanzania	463	Tanzania health centers data	EBF, WHO definition (interview)	Community-acquired pneumonia	Lack of EBF during the first 6 months was associated with higher risk of pneumonia (OR = 1.7; 95% CI 1.0, 2.9)
Rustam et al. (2019) [49]	Indonesia	324	Kampar District primary health care data	EBF, no definition (questionnaire)	Upper respiratory infection	Non-EBF group had increased odds of upper respiratory infection EBF group (OR = 1.69; 95% CI 1.02, 2.80).
Sutriana et al. (2021) [50]	Indonesia	176	Bojonegoro Regency health center data	EBF, during 0 to 6 months, no definition (questionnaire)	Pneumonia	Non-EBF group had increased odds of pneumonia than EBF group (OR = 7.95; 95% CI 3.52, 17.94).
Yadav et al. (2013) [51]	Nepal	200	College of Medical Sciences Teaching Hospital data	EBF, no definition (recorded in proforma)	ARI	Difference between ARI in case and control group for EBF is not statistically significantly different (OR = 1.33; 95% CI 0.73, 2.44; *p* = 0.36).
**Case–control studies (developed countries)**						
Pandolfi et al. (2017) [52]	Italy	296	Italian pediatric hospital data	EBF, no definition (questionnaire)	Pertussis	EBF was not statistically significantly associated with pertussis compared to partial BF and artificial feeding (OR = 1.2; 95% CI 0.31, 4.67; *p* = 0.779)
**Cross-sectional studies (developing countries)**						
Abdulla et al. (2022) [53]	Bangladesh	5724	BDHS data	EBF (babies currently breastfed and no complementary foods) 24 h recall	Diarrhea, ARI, combination	Non-EBF group had higher risk of diarrhea (OR = 2.11; 95% CI 1.56, 2.85), acute respiratory infection (OR = 1.43; 95% CI 1.28, 1.60) or both (OR = 1.48; 95% CI 1.32, 1.66) compared to EBF group.
Adhiningsih et al. (2020) [54]	Indonesia	116	Primary health care records and standard questionnaire	History of EBF	Rotavirus infection	EBF group were less likely to have rotavirus diarrhea (OR = 0.67; 95% CI 0.26, 1.75) when compared to non-EBF group (Not statistically significantly different (*p* > 0.05)).
Ahmed et al. (2020) [55]	Ethiopia	15,106	EDHS data	EBF (24 h recall)	Diarrhea, ARI	EBF group were lower odds of diarrhea (OR = 0.51; 95% CI 0.39, 0.65). EBF group had lower odds of ARI (OR = 0.65; 95% CI 0.51, 0.83)
Cai et al. (2016) [56]	China	1654	Maternal Infant Nutrition and Growth (MING) study	EBF, breast milk as sole source of milk, with or without water, supplements, medication and food since birth (questionnaire)	Diarrhea, respiratory (bronchitis, cold, pneumonia)	Compared to EBF group, exclusive formula-feeding group had higher odds of respiratory illness (AOR = 1.44; 95% CI 1.04, 2.00). Compared to EBF group, mixed feeding group had higher odds of diarrhea (OR = 1.40; 95% CI 1.00, 1.96).
Dagvadorj et al. (2016) [57]	Mongolia	1083	Bulgan hospital data	EBF (no definition) (questionnaire)	Lower respiratory tract infection (LRTI)	EBF for >4 months was found to be a negative predictor of children’s hospitalization for LRTI (AOR = 0.43; 95% CI 0.24, 0.74).
Dairo et al. (2017) [58]	Nigeria	630	Primary health care centre data	EBF (no definition)	Diarrhea	Compared to EBF group, babies who got partial BF (OR = 4.59; CI 3.20, 6.99) and predominant BF (OR = 2.78; CI 1.79, 4.29) had higher odds of diarrhea.
Demissie et al. (2021) [59]	Ethiopia	414	Wolaita Sodo University Teaching and Referral Hospital data	EBF, only breast milk for first 6 months or only breast milk until assessment time for babies aged < 6 months. (questionnaire)	Acute lower respiratory tract infection	Non-EBF group more likely to experience acute lower respiratory tract infections relative to EBF group (AOR = 1.85; 95 % CI 1.18, 2.91, *p* = 0.008).
Dhami et al. (2020) [60]	India	90,596	National Family Health Survey (NFHS-4)/India (DHS)	EBF (WHO definition) (24 h recall survey)	Diarrhea	EBF group aged 0–5 months had a lower risk of diarrhea in areas of North, East, and Central India, as well as nationally (AOR = 0.64; 95% CI 0.57, 0.72) compared to their counterparts.
Flores et al. (2021) [61]	Peru	279	The Parent Study data	EBF (WHO definition) (questionnaire)	Tuberculosis	Children who were EBF for 6 months had a greater prevalence (PR = 1.6; 95% CI = 0.9, 2.7) of active tuberculosis compared to non-EBF group (not statistically significantly different (*p* > 0.05).
Gizaw et al. (2017) [62]	Ethiopia	367	Hadaleala district study data	EBF, no definition (questionnaire)	Diarrhea	Odds of diarrhea was higher in infants aged <6 months and infants aged 6 to 24 months in non-EBF group during first 6 months (AOR = 13.33; 95% CI 1.59, 112.12) (AOR = 19.24; 95% CI 8.26, 44.82) compared to EBF group.
Hajeebhoy et al. (2014) [63]	Vietnam	6068	Alive & Thrive (A&T) program data	EBF (WHO definition) 24 h recall questionnaire)	Diarrhea, ARI	Compared to EBF group, babies predominantly (AOR = 1.52; 95% CI 1.05, 2.21) and partially breastfed (AOR = 1.55; 95% CI 1.07, 2.24) had higher odds of diarrhea. Compared to EBF group, babies partially breastfed (AOR = 1.24; 95% CI 1.03, 1.48) had higher odds of ARI.
Issa et al. (2019) [64]	Lebanon	222	Beirut study data	EBF (WHO definition) (questionnaire)	Urinary tract infection (UTI)	Infants who were EBF for a longer duration had lower odds of UTIs compared to infants who were not EBF for a longer duration (OR = 0.77; 95% CI 0.63, 0.95).
Kembo et al. (2011) [65]	Zimbabwe	3220	ZDHS data	EBF (24 h recall survey)	Diarrhea	EBF group had a lower risk of diarrhea compared to non-EBF group (OR = 0.27; 95% CI 0.09, 0.74)
Khan et al. (2017) [66]	Bangladesh	1918	BDHS data	EBF (24 h recall survey)	ARI, diarrhea, fever	Odds of having diarrhea, ARI, and fever was greater if EBF was discontinued during 0 to 2 months (OR = 4.94; 95% CI 3.17, 10.23) (OR = 2.38; 95% CI 1.27, 3.26) (OR = 2.18; 95% CI 1.56, 3.04), 2 to 4 months (OR = 3.07; 95% CI 2.11, 5.03) (OR = 1.40; 95% CI 1.10, 1.76) (OR = 1.53; 95% CI 1.37, 2.10), and 4 to 6 months (OR = 2.30; 95% CI 1.89, 3.20) (OR = 1.19; 95% CI 1.04, 1.57) (OR = 1.23; 95% CI 1.06, 1.63) compared to infants who did not stop EBF up to 6 months.
Mulatu et al. (2021) [67]	Ethiopia	1034	EDHS data	EBF (24 h recall survey)	Diarrhea, fever	EBF group had reduced odds of diarrhea (OR = 0.33; 95% CI 0.13, 0.83) and fever (OR = 0.34; 95% CI 0.16, 0.75) relative to non-EBF group.
Mulatya et al. (2020) [68]	Kenya	20,964	KDHS data	EBF (24 h recall survey)	Diarrhea, ARI comorbidity	EBF did not have significant association with diarrhea and ARI comorbidity (OR = 2.73; 95% CI 0.80, 9.28; *p* = 0.107)
Nigatu et al. (2019) [69]	Ethiopia	2433	EDHS data	EBF, only breast milk in the 24 h before questionnaire	ARI, diarrhea, fever	Stopping EBF at 3 months (OR = 1.95; 95% CI 1.08, 3.53) and 4–6 months (OR = 3.5; 95% CI 2.19, 5.83) raised the odds of diarrhea compared to infants who were EBF for 6 months. Stopping EBF at 4–6 months raised the odds of ARI (OR = 2.74; 95% CI 1.61, 4.65) and fever (OR = 1.73; 95% CI 1.11, 2.68).
Ogbo et al. (2017) [70]	Uganda, Burkina Faso, Ethiopia, Mali, Nigeria, DR Congo, Tanzania, Niger, and Kenya	83,100	DHS data	EBF (aged 0–5 months), only breast milk (medicines, vitamins, ORS acceptable)	Diarrhea	EBF was associated with decreased risk of diarrhea (OR = 0.50; 95% CI 0.43, 0.57).
Ogbo et al. (2018) [71]	Tanzania	10,139	TDHS data	EBF (aged 0–5 months), only breast milk (medicines, vitamins, ORS acceptable)	Diarrhea	EBF group had reduced odds of diarrhea compared to non-EBF group (OR = 0.31; 95% CI 0.16, 0.59).
Ogbo et al. (2016) [72]	Nigeria	88,152	NDHS data	EBF (aged 0–5 months), only breast milk (medicines, vitamins, ORS acceptable)	Diarrhea	BEBF group had reduced odds of diarrhea compared to non-EBF group (RR = 0.61; 95% CI 0.44, 0.86).
Puspitasari et al. (2021) [73]	Indonesia	16,555	IDHS data	EBF (aged 0–6 months), with no complimentary foods	ARI	EBF group had reduced odds of ARI compared to non-EBF group (OR = 0.85; 95% CI 0.73, 0.99).
Saeed et al. (2020) [74]	Pakistan	1033	PDHS data	EBF (survey)	ARI, diarrhea, fever	EBF group had reduced odds of ARI (OR = 0.53; 95% CI 0.32, 0.90), fever (OR = 0.72; 95% CI 0.52, 0.99), and diarrhea (OR = 0.66; 95% CI 0.45, 0.98).
Santos et al. (2016) [75]	Brazil	854	Family Health Strategy data	EBF, only breast milk, WHO definition (interview)	Diarrhea	Babies aged <6 months who were EBF had lower odds of diarrhea than babies who were mixed BF (OR = 10.8; 95% CI 2.3, 49.6) and complementary BF (OR = 14.1; 95% CI 3.3, 60.3).
Shumetie et al. (2018) [76]	Ethiopia	553	Bahir Dar study data	EBF, no definition (questionnaire)	Diarrhea	Non-EBF group had increased odds of diarrhea compared to EBF group (OR = 2.69; 95% CI 1.39, 5.19).
Srivastava et al. (2020) [77]	India	94,144	(NFHS-4) data	EBF, only breast milk (medication, vitamins, ORS, minerals acceptable) (aged 0–5 months)	Diarrhea, ARI	EBF group had reduced odds of diarrhea (OR = 0.78; 95% CI 0.72, 0.84; *p* < 0.01) and ARI (OR = 0.99; 95% CI 0.85, 1.15) than non-EBF group (result for ARI statistically significantly different (*p* > 0.1)).
Wibowo et al. (2021) [78]	Indonesia	13,921	IDHS data	EBF, no definition (questionnaire)	Diarrhea	EBF group were more likely to not experience diarrhea than non-EBF group (OR = 1.71; 95% CI 1.29, 1.07).
**Cross-sectional studies (developed countries)**						
Alexandrino et al. (2016) [79]	Portugal	152	Porto day care center data	EBF (no definition)	Lower respiratory tract infection	Risk of lower respiratory tract infection was greater in non-EBF group compared to EBF group (OR = 24.61; 95% CI 1.11, 546.53).
Payne et al. (2017) [80]	UK	30,760	Infant Feeding Survey (IFS) data	EBF, WHO definition (questionnaire)	Hospitalization for infectious causes	Among babies breastfed at least 6 months, those who were EBF for 6 weeks or more had a lower risk for infectious causes (IR = 0.48; 95% CI 0.32, 0.72) compared to those not EBF for 6 weeks (IR = 0.72; 95% CI 0.48, 1.08).

ARI—acute respiratory infection; BF—breastfeeding; CI—confidence interval; DHS—demographic health survey; ETEC—enterotoxigenic *Escherichia coli*; GI—gastrointestinal; HR—hazard ratio; ORS—oral rehydration solution; OR—odds ratio; PR—prevalence ratio; RR—risk ratio.

## Data Availability

Not applicable.

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
