# Peer review of "Exclusive Breastfeeding and Childhood Morbidity: A Narrative Review"

_ijerph, 2022, doi:10.3390/ijerph192214804_

Round 1

Reviewer 1 Report

This narrative review of exclusive breastfeeding and infectious disease in under-5-year-old children is well-researched and generally well-written. The authors explain the difficulties in interpreting the findings  (definitional issues, etc.) and draw reasonable conclusions. Their mention of possible mechanisms underlying the protective effect of breastfeeding was appropriate. They could consider mentioning the protective effect of EBF on other childhood conditions such as sudden infant death syndrome. Here the work of McKenna and others is worth a mention. Much of the breastfeeding work in SIDS seems to support the idea that infection plays a key role.

Some specific points... P1 line 43 ... I suggest changing "due to " to: "has been attributed to"

Line 43... The sentence is missing some words; '...aged less than in LMICS"  ??

Line 234... change "reports" to "report"

Author Response

Response to Reviewer 1 Comments

We thank the reviewer for his/her valuable suggestions. We addressed the comments below:

Point 1: This narrative review of exclusive breastfeeding and infectious disease in under-5-year-old children is well-researched and generally well-written. The authors explain the difficulties in interpreting the findings  (definitional issues, etc.) and draw reasonable conclusions. Their mention of possible mechanisms underlying the protective effect of breastfeeding was appropriate. They could consider mentioning the protective effect of EBF on other childhood conditions such as sudden infant death syndrome. Here the work of McKenna and others is worth a mention. Much of the breastfeeding work in SIDS seems to support the idea that infection plays a key role.

 Response 1: Yes, we agree with this suggestion. We have now included a paragraph in the discussion section addressing benefits of EBF on reducing the risk of sudden infant death syndrome (Line 1087-1093 of the revised manuscript). Two studies (One by McKenna and a meta-analysis) have been added on the protective effects of breastfeeding against SIDS [93,94]. The meta-analysis also discusses a few plausible biological mechanisms for this effect such as the immunological properties of breastmilk which may particularly safeguard babies during the vulnerable timeframe (2-4 months) when SIDS are more likely to occur. Thank you for providing the relevant reference.

Point 2: Some specific points... P1 line 43 ... I suggest changing "due to " to: "has been attributed to"

Response 2: Line 43 of the original manuscript has been changed from “due to” to “have been attributed to”.

Point 3: Line 43... The sentence is missing some words; '...aged less than in LMICS"  ??

Response 3: We apologize for not making the sentence clear. The missing words “six months” has now been added to Line 50 of the revised manuscript.

Point 4: Line 234... change "reports" to "report"

Response 4: We were not able to find the word “reports” in Line 234. However, we found that we made a similar error in Line 234 of the original manuscript and have changed “support” to “supports”. 

Reviewer 2 Report

This manuscript reviews the effect of exclusive breastfeeding on the incidence of gastrointestinal and respiratory infections. The manuscript's structure is fine, the content is understandable and convincing, and it is worthy of publication.

The reviewers would like to see the paper revised as some of the contents are difficult to read.

1) Regarding Table 2, is it unnecessary to include detailed main results? It is sufficient to know whether the drugs are prophylactic or not for gastrointestinal infections and whether they are prophylactic or not for respiratory infections. Since reference numbers are cited, the reader can search the literature if necessary.

2) For results L122 - L151, it is preferable to make a new table and put the contents of the text in a footnote or in the text as a description of the new table. It is easier to read that way.

3) The conclusion is only one sentence in L232, and L232-L246) should be moved to the discussion section.

Author Response

Response to Reviewer 2 Comments

We thank the reviewer for his/her valuable suggestions. We addressed the comments below:

This manuscript reviews the effect of exclusive breastfeeding on the incidence of gastrointestinal and respiratory infections. The manuscript's structure is fine, the content is understandable and convincing, and it is worthy of publication.

Response: Thank you.

The reviewers would like to see the paper revised as some of the contents are difficult to read.

Point 1: Regarding Table 2, is it unnecessary to include detailed main results? It is sufficient to know whether the drugs are prophylactic or not for gastrointestinal infections and whether they are prophylactic or not for respiratory infections. Since reference numbers are cited, the reader can search the literature if necessary.

Response 1: Many thanks for your helpful comment. We have now edited and reduced many unnecessary words in the main results and outcome variables column of Table 2 for the studies. We think it may be necessary to include this level of detail as it adds depth and it is also easier and less time-consuming for the readers to compare rather than search the references.

Point 2: For results L122 - L151, it is preferable to make a new table and put the contents of the text in a footnote or in the text as a description of the new table. It is easier to read that way.

Response 2: Thank you for your suggestion. We did revise and intend to include a table but it was very short so we decided to leave the text here as we think there is not the detail needed for a table. We are happy to add the table back if the editor requests it. Thanks again for this suggestion.

Point 3: The conclusion is only one sentence in L232, and L232-L246) should be moved to the discussion section.

Response 3: Thank you, we agree with this suggestion. Only the first and the last sentence of this paragraph now remains as the conclusion and the rest of the paragraph has been moved to the discussion section (Line 1045-1053) of the revised manuscript). Any new information that we introduced in the conclusion previously has now been removed.

Reviewer 3 Report

Not aged 0 – from birth

Define LMICS at first use

Line 34 – that is a very specific number of deaths that is limited by that reference of course, I would round up or down and approximate

Table 1 – on what basis were papers excluded if sample size was less than 100? How was power calculated?

Figure 1 – improve readability by using borders and colours, consult PRISMA flowchart

Table 2 – consider changing to landscape to make it easier to read ; also legend needs expanding to explain to the reader what parameters are included in the table

Bacteria names should be in italics  

Overall a useful and well written review , but maybe a section on what do children that are not exclusively breastfed consume? Is there any particular formula milk that is better or worse? May be beyond scope of paper but a brief mention might be useful

Perhaps a section of components of breast milk vs formula milk – why is it better? Perhaps an original figure to explain difference, like BioRender schematics

Numbers under 10 should be in words

You shouldn’t introduce new concepts in the conclusion – what is the WHO code and the Baby friendly hospital initiative guidelines? Either remove or add a few lines on them in intro to explain them to reader

Ref 37 and 85, year not in bold, for consistency

Author Response

Response to Reviewer 3 Comments

We thank the reviewer for his/her valuable suggestions. We addressed the comments below:

Point 1: Not aged 0 – from birth

Response 1: This has now been corrected and “from birth” replaces “aged 0” (Line 12 of revised version of manuscript).

Point 2: Define LMICS at first use

Response 2: We apologize for not making that definition clear. LMICs has now been defined in Line 50 of the revised version of the manuscript.

Point 3: Line 34 – that is a very specific number of deaths that is limited by that reference of course, I would round up or down and approximate

Response 3: Yes, we agree with that suggestion. We have rounded up and approximated “652,572 deaths” to “almost 653,000 deaths” in Line 34 of the revised version of the manuscript.

Point 4: Table 1 – on what basis were papers excluded if sample size was less than 100? How was power calculated?

Response 4: We aimed to include studies with large sample size over a 100 mother-infant pairs. We did not calculate power as we left it up to the individual studies. We believe the power would have been limited due to the sample size and there may have not been an ability to see a difference if the sample size was small.

Point 5: Figure 1 – improve readability by using borders and colours, consult PRISMA flowchart

Response 5: We agree with this suggestion and formatted figure 1 according to PRISMA flowchart.

Point 6: Table 2 – consider changing to landscape to make it easier to read ; also legend needs expanding to explain to the reader what parameters are included in the table

Response 6: We agree with this comment. Table 2 is now changed to landscape. We have also edited the legend to “Summary table of studies investigating the association between EBF and childhood infections arranged according to study design and country type (n=70)”.

Point 7: Bacteria names should be in italics  

Response 7: All bacteria names in the revised manuscript have now been typed in italics (Table 2 and Lines 977 and 1067) .

Point 8: Overall a useful and well written review , but maybe a section on what do children that are not exclusively breastfed consume? Is there any particular formula milk that is better or worse? May be beyond scope of paper but a brief mention might be useful

Response 8: Thank you. We believe that this is beyond the scope of this paper. However, there are other reviews in this journal that cover this in different contexts.

Point 9: Perhaps a section of components of breast milk vs formula milk – why is it better? Perhaps an original figure to explain difference, like BioRender schematics

Response 9: Thank you, we agree with this suggestion. We have outlined some of the beneficial components of breastmilk (Line 1056-1059 of revised manuscript). We have also added some components that are unique to breastmilk such as antibodies, hormones, and enzymes (Line 1059-1062).

Point 10: Numbers under 10 should be in words

Response 10: Thank you for correcting this. All numbers under 10 are now in words.

Point 11: You shouldn’t introduce new concepts in the conclusion – what is the WHO code and the Baby friendly hospital initiative guidelines? Either remove or add a few lines on them in intro to explain them to reader

Response 11: We agree and have removed any new concepts from the conclusion and added them into the discussion section. (Line 1045-1053 of revised manuscript). We have also removed ‘the WHO code and the Baby friendly hospital initiative guidelines’ statement.

Point 12: Ref 37 and 85, year not in bold, for consistency

Response 12: We have formatted the font to bold for the year in reference number 37 and 85.
